# Three New Species of *Apiospora* (Amphisphaeriales, Apiosporaceae) on *Indocalamus longiauritus*, *Adinandra glischroloma* and *Machilus nanmu* from Hainan and Fujian, China

**DOI:** 10.3390/jof10010074

**Published:** 2024-01-17

**Authors:** Xinye Liu, Zhaoxue Zhang, Shi Wang, Xiuguo Zhang

**Affiliations:** 1College of Life Sciences, Shandong Normal University, Jinan 250358, China; line0524@126.com (X.L.); wangssdau@126.com (S.W.); 2Shandong Provincial Key Laboratory for Biology of Vegetable Diseases and Insect Pests, College of Plant Protection, Shandong Agricultural University, Taian 271018, China; zhangzhaoxue2022@126.com

**Keywords:** taxonomy, *Ascomycota*, multigene phylogeny

## Abstract

*Apiospora* is widely distributed throughout the world, and most of its hosts are *Poaceae*. In this study, *Arthrinium*-like strains were isolated from non-*Poaceae* in the Hainan and Fujian provinces of China. Based on the combined DNA sequence data of the internal transcriptional spacer (ITS), partial large subunit nuclear rDNA (LSU), translation extension factor 1-α gene (*TEF1-α*) and β-tubulin (*TUB2*), the collected *Apiospora* specimens were compared with known species, and three new species were identified. Based on morphological and molecular phylogenetic analyses, *Apiospora adinandrae* sp. nov., *A. bawanglingensis* sp. nov. and *A. machili* sp. nov. are described and illustrated.

## 1. Introduction

The genus *Apiospora* (*Apiosporaceae*, *Amphisphaeriales*, *Sordariomycetes*, *Ascomycota*) was identified and established by Saccardo (1875) with *Apiospora montagnei* as the type species [1]. The *Arthrium*-like taxa under *Apiosporaceae* contains the genus with apiosporous hyaline ascospores and basauxic conidiogenesis [2,3]. Thus, after Ellis (1965), the biological relationship between *Apiospora* and *Arthrinium* became widely accepted, and *Apiospora* was considered a synonym for *Arthrinium* until the study of Pintos and Alvarado, the former was usually considered the sexual morph and the latter was considered the asexual morph [3,4,5]. During this time, the dual nomenclature was abandoned, and the old name *Arthrinium* was recommended for unitary nomenclature [6].

The genus *Arthrinium* was introduced by Kunze and Schmidt (1817) with *Ar. caricicola* as the type species [7]. In order to clarify the exact status of the *Arthrium*-like taxa, most species were recollected, and some species were reassigned to different genera by Link (in Willdenow 1824) due to different conidial shapes, but were subsequently reclassified back to *Arthrinium* by von Höhnel 1925 and Cooke 1954, because of similar conidiophores [8,9,10]. Pintos et al. (2019) published the genetic information of the first model species *Ar. caricicola*, followed by the phylogenetic analysis of Pintos and Alvarado (2021), which found that *Arthrinium* formed a separate clade from other sequences of *Apiospora*, and *Apiospora* and *Arthrinium* were phylogenetically distinguished [4,11].

The morphological similarities between the two genera (*Apiospora* and *Arthrinium*) make it difficult to determine the boundary between them by morphological methods alone. Morphologically, most of the conidia of *Apiospora* are nearly spherical in the front and lenticular at the side, and sometimes the conidia develop into pinholes, and the conidia of *Arthrinium* vary in shape (angular, spherical, curved, boat-shaped, fusiform, polygonal), with some species having thick black intervals [12,13,14,15].

Ecologically, *Apiospora* was widely distributed in various climatic zones around the world and was found in many species of hosts, whereas *Arthrinium* was rarely distributed in tropical and subtropical regions and had fewer hosts [13,14]. Species of *Apiospora* have been found in *Poaceae* during past studies, while *Arthrinium* has been found in the family *Cyperaceae* and *Juncaceae* [4,15,16,17,18,19,20]. The ecological differences further support the genetic and morphological division of the two genera, and all the evidence is sufficient to support the separation of the two genera in taxonomic status. In addition, the host association and geographical distribution were discussed based on the existing research progress.

Currently, there are 120 records of *Arthrinium* and 157 records of *Apiospora* on Index Fungorum (http://www.indexfungorum.org/, accessed on 20 November 2023). In the present study, the authors isolated strains of *Arthrinium*-like taxa in China. To clarify the taxonomic status, morphological and phylogenetic studies were conducted. Three new species, *Apiospora adinandrae* sp. nov., *A. bawanglingensis* sp. nov. and *A. machili* sp. nov., were identified and classified in *Apiospora* by multilocus analysis of tandem internal transcribed spacer (ITS), 28S large subunit ribosomal RNA gene (LSU), translation elongation factor 1-alpha gene (*TEF1α*) and beta-tubulin (*TUB*) datasets.

## 2. Materials and Methods

### 2.1. Isolation and Morphology

#### 2.1.1. Strains Isolation

The four strains were isolated from three host plants in two provinces of China (Table 1). These strains were obtained by tissue separation and the single spore isolation method [21]. Surface sterilization was performed on 5 × 5 mm fragments at the junction of the leaf lesion edge and healthy tissue. The fragments were soaked in 75% alcohol for 90 s, washed with sterile water for 45 s, then immersed in 5% sodium hypochlorite solution for 60 s and finally washed with sterile water 3 times. The sterilized wet fragments were dried on sterilized filter paper and then cultured on potato dextrose agar (PDA: 220 g potato, 20 g agar, 18 g dextrose, 1000 mL sterile water and natural pH) 25 °C for 3 days. Subsequently, morphologically different colonies could be observed on PDA, and the tip of a hyphae with strong growth capacity growing on the edge of the colony was picked and transferred to another Petri dish containing PDA and culture was continued at 25 °C.

#### 2.1.2. Morphological Studies

The colonies were observed morphologically on the 7th and 14th day of culture and photographed by digital camera (Canon Powershot G7X, Canon, Tokyo, Japan). A stereomicroscope (Olympus SZX10, OLYMPUS, Tokyo, Japan) and a microscope (Olympus BX53, OLYMPUS, Tokyo, Japan) were used to observe the microscopic morphological characteristics of the colonies. Both microscopes were equipped with high-definition color digital cameras to capture the conidia of fungal structures.

In order to facilitate further research, the four strains in this study were kept in 10% sterilized glycerol or sterile water at 4 °C. Voucher specimens were stowed in two institutions, the Herbarium of the Department of Plant Pathology (HSAUP) of Shandong Agricultural University, Taian, China, and the Herbarium Mycologicum Academiae Sinicae (HMAS) of the Institute of Microbiology, Chinese Academy of Sciences, Beijing, China. Ex-holotype living cultures were stored in the Shandong Agricultural University Culture Collection (SAUCC). All taxonomic information on the new taxa obtained in this paper has been submitted to MycoBank (http://www.mycobank.org, accessed on 20 November 2023).

### 2.2. DNA Extraction and Amplification

Fungal tissue was obtained on PDA and genomic DNA was extracted from the mycelium. DNA extraction was performed by a magnetic bead kit (OGPLF-400, GeneOnBio Corporation, Changchun, China) and the CTAB method [22,23]. The polymerase chain reaction (PCR) procedure was performed using the primer pairs in Table 2, which contains the entire internal transcriptional spacer (ITS) of the intervening 5.8S rRNA gene, part of the large subunit nrRNA gene (LSU), part of the translation extension factor 1-α gene (*TEF1α*) and part of the beta-tubulin gene (*TUB2*).

The polymerase chain reaction was carried out using an Eppendorf Master Thermocycler (Hamburg, Germany). The amplification reaction was performed in a 20 μL reaction system consisting of 10 μL 2 × Hieff Canace^®^ Plus PCR Master Mix (With Dye) (Yeasen Biotechnology, Cat No. 10154ES03, Shanghai, China), the forward and reverse primer (TsingKe, Qingdao, China) each 0.7 μL per 10 μM, and 1.4 μL template genomic DNA, at last replenishing the total volume to 20 μL with distilled deionized water. The PCR products were separated by 1% agarose gel electrophoresis with GelRed added, and the bands were observed under ultraviolet light [24]. Then, we used a Gel Extraction Kit (Cat: AE0101-C) (Shandong Sparkjade Biotechnology Co., Ltd., Ji’nan, China) for gel recovery. The bidirectional sequencing of DNA samples was completed by the Tsingke Company Limited (Qingdao, China), and the resulting sequences were processed by MEGA 7.0 to achieve consistency [25]. The sequences data obtained and used in this article were all uploaded to GenBank (Appendix A).

**Table 2 jof-10-00074-t002:** Molecular markers and their PCR primers and programs used in this study.

Locus	Primers	Sequence (5′–3′)	PCR Cycles	References
ITS	ITS5	GGA AGT AAA AGT CGT AAC AAG G	(95 °C: 30 s, 55 °C: 30 s, 72 °C: 45 s) × 29 cycles	[26]
	ITS4	TCC TCC GCT TAT TGA TAT GC		
LSU	LR0R	GTA CCC GCT GAA CTT AAG C	(95 °C: 30 s, 48 °C: 50 s, 72 °C: 1 min 30s) × 35 cycles	[27]
	LR5	TCC TGA GGG AAA CTT CG		
*TEF1α*	EF1-728F	CAT CGA GAA GTT CGA GAA GG	(95 °C: 30 s, 51 °C: 30 s, 72 °C: 1 min) × 35 cycles	[28]
	EF2	GGA RGT ACC AGT SAT CAT GTT		
*TUB2*	T1	AAC ATG CGT GAG ATT GTA AGT	(95 °C: 30 s, 56 °C: 30 s, 72 °C: 1 min) × 35 cycles	[29]
	Bt-2b	ACC CTC AGT GTA GTG ACC CTT GGC		

### 2.3. Phylogenetic Analyses

The new sequences obtained in present study were all compared in NCBI’s Genbank nucleotide database (https://www.ncbi.nlm.nih.gov/, accessed on 11 November 2023) and all reference sequences for relevant species were downloaded. Multi-sequence analysis was performed using MAFFT 7 online services and the auto policy (http://mafft.cbrc.jp/alignment/server/, accessed on 20 November 2023) to compare the newly generated sequences with other related sequences. To identify isolates at the species level, a phylogenetic analysis of each marker was first performed and then combined (ITS-LSU-*TEF1α*-*TUB2*) (See Appendix A).

Phylogenetic analysis of multi-labeled data was performed based on Bayesian inference (BI) and maximum likelihood (ML) algorithms. Both ML and BI were run on the CIPRES Science Gateway portal (https://www.phylo.org/, accessed on 20 November 2023) or offline software (The ML was operated in RaxML-HPC2 on XSEDE v8.2.12 and BI analysis was operated in MrBayes v3.2.7a with 64 threads on Linux) [30,31,32,33,34,35]. For ML analyses, the default parameters were used and 1000 rapid bootstrap replicates were run with the GTR+G+I model of nucleotide evolution, and the BI analysis was performed using a fast bootstrap algorithm with an automatic stop option. The BI analyses included 2 million generations of sixty-four parallel threads with the stop rule options and 100 generations of sampling frequency. The burn-in score was set to 25% and posterior probabilities (PP) were determined from the remaining trees. All resulting trees were plotted using FigTree v. 1.4.4 (http://tree.bio.ed.ac.uk/software/figtree, accessed on 20 November 2023), or ITOL: Interactive Tree Of Life (https://itol.embl.de/, accessed on 21 November 2023) [36], and the layout of the trees was produced in Adobe Illustrator CC 2019.

## 3. Results

### 3.1. Phylogenetic Analyses

Phylogenetic analysis was performed on 99 isolates representing *Apiospora* species, of which 97 isolates were considered to be the ingroup and 2 strains of *Arthrinium caricicola* (CBS 145127) were used as the outgroup. The final alignment contained 2843 concatenated characters, viz. 1–838 (ITS), 839–1674 (LSU), 1675–2293 (*TEF1α*), 2294–2843 (*TUB2*), 1619 were constant, 421 were variable and parsimony-uninformative and 803 were parsimony-informative. The topology of the ML tree confirms the tree topology obtained from Bayesian inference; therefore, only the ML tree is presented (Figure 1). The alignment has 1456 distinct alignment patterns. The proportion of gaps and completely undetermined characters in this alignment: 33.53%. The estimated base frequencies were as follows: A = 0.234029, C = 0.250395, G = 0.257923, T = 0.257653; substitution rates AC = 1.257520, AG = 3.156093, AT = 1.159778, CG = 0.983356, CT = 4.511962 and GT = 1.000000; gamma distribution shape parameter α = 0.263195. Final ML optimization likelihood: −25299.135520. The GTR+I+G model was proposed for ITS, LSU, *TEF1α* and *TUB2*. BI analysis of these four tandem genes was performed over 3,400,000 generations in 68,002 trees. The first 17,000 trees representing the burn-in phase of the analysis are discarded, while the remaining trees are used to calculate the posterior probability in the majority rule consensus tree (Figure 1; first value: PP > 0.90 shown). The alignment embodied a total of 1469 unique site patterns (ITS: 475, LSU: 204, *TEF1α*: 440, *TUB2*: 350).

In our phylogenetic analyses, 97 strains of *Apiospora* were identified as a monophyletic clade (Figure 1). Among them, 8 strains produced three new species lineages, *Apiospora adinandrae* (SAUCC 1282B-1, SAUCC 1282B-2) closely related to *A. aurea* (CBS 244.83), *A. hydei* (CBS 114990) and *A. cordylines* (GUCC 10027) with full support (99% MLBV and 1.0 BIPP); *A. bawanglingensis* (SAUCC BW0444 and SAUCC BW04441) closely related to *A. piptatheri* (CBS 145149) with good support (1.0 BIPP and 76% MLBV); and *A. machili* (SAUCC 1175A-4) forming a separate single-species lineage. The present study revealed three species, viz. *Apiospora adinandrae* sp. nov., *A. bawanglingensis* sp. nov. and *A. machili* sp. nov.

### 3.2. Taxonomy

#### 3.2.1. *Apiospora adinandrae* X.Y. Liu, Z.X. Zhang and X.G. Zhang, sp. nov. (Figure 2)

MycoBank—No: 850667;Etymology—The epithet *adinandrae* pertains to the generic name of the host plant *Adinandra glischroloma*;Type—Wuyishan National Forest Park, Fujian Province, China, on diseased leaves of *Adinandra glischroloma*, 15 October 2022, X.Y. Liu (HMAS 352657, holotype), ex-holotype living culture SAUCC 1282B–1;Description—On PDA, hyphae 2.5–4.0 μm in diameter, branched, hyaline and septate. Sexual morph: Undetermined. Asexual morph: Conidiophore reduced to conidiogenous cells, aggregated in clusters on hyphae. Conidiogenous cells are 3.5–6.0 × 2.0–3.5 μm, hyaline becoming pale green, polyblastic, cylindrical, septate, verrucose, flexuous. Conidia smooth, rounded to ovoid, globose to subglobose, green to pale brown, 7.5–12.4 × 6.3–10.5 μm, mean ± SD = 9.1 ± 1.0 × 8.1 ± 1.1 μm. See Figure 2;Culture characteristics—PDA, colonies concentrically spreading with subcircular margin, flat, abundant white aerial mycelium. In reverse, the sites where mycelium is abundant appear pale yellow, and the sites with sparse mycelium appear yellow. After seven days of incubation at 25 °C, the colony diameter reached 57.5–65.5 mm and the growth rate was 8.21–9.35 mm/day;Additional specimen examined—China, Fujian Province: Wuyishan National Forest Park, on diseased leaves of *Adinandra glischroloma*, 15 October 2022, X.Y. Liu, HSAUP 1282B–2, living culture SAUCC 1282B–2;Notes—Phylogenetic analyses based on ITS-LSU-*TEF1α*-*TUB2* rDNA sequences showed that *Apiospora adinandrae* sp. nov. formed an independent clade which is closely related to *A. aurea* (CBS 244.83), *A. hydei* (CBS 114990) and *A. cordylines* (GUCC 10027). The base-pair comparison of ITS, LSU, *TEF1α* and *TUB2* sequences, respectively, showed 1.01%, 0.48%, 6.81% and 3.62% differences between *A. adinandrae* (SAUCC 1282B-1) and *A. aurea* (CBS 244.83); showed 1.18%, 0.36%, 6.79% and 4.75% differences between *A. adinandrae* and *A. hydei* (CBS 114990); and showed 1.04%, 0%, 6.47% and 4.97% differences between *A. adinandrae* and *A. cordylines* (GUCC 10027);Morphologically, *A. adinandrae* differs from *A. aurea*, *A. hydei* and *A. cordylines* in conidiophore, conidiogenous cells and conidia. The conidiophore of *A. adinandrae* usually reduced to conidiogenous cells, while both *A. aurea* and *A. cordylines* have brown transverse septa conidiophore. The conidiogenous cells of *A. adinandrae* hyaline become pale green, while the conidiogenous cells of *A. aurea*, *A. hydei* and *A. cordylines* become colorless to brown. *A. aurea* has dark brown conidia (10.0–30.0 × 10.0–15.0 μm) and sterile cells of a different shape than the conidia. The conidia of *A. hydei* are brown, roughened, globose in surface view (15.0–22.0 × 10.0–14.0 μm). The conidia of *A. cordylines* brown, smooth to finely roughened, subglobose (15.0–19.0 × 12.5–18.5 μm). The conidia of *A. adinandrae* are smaller in size (7.5–12.4 × 6.3–10.5 μm) and more oval in shape than the conidia of *A. aurea*, *A. hydei* and *A. cordylines* [4]. For details, see Table 3.

**Figure 2 jof-10-00074-f002:**
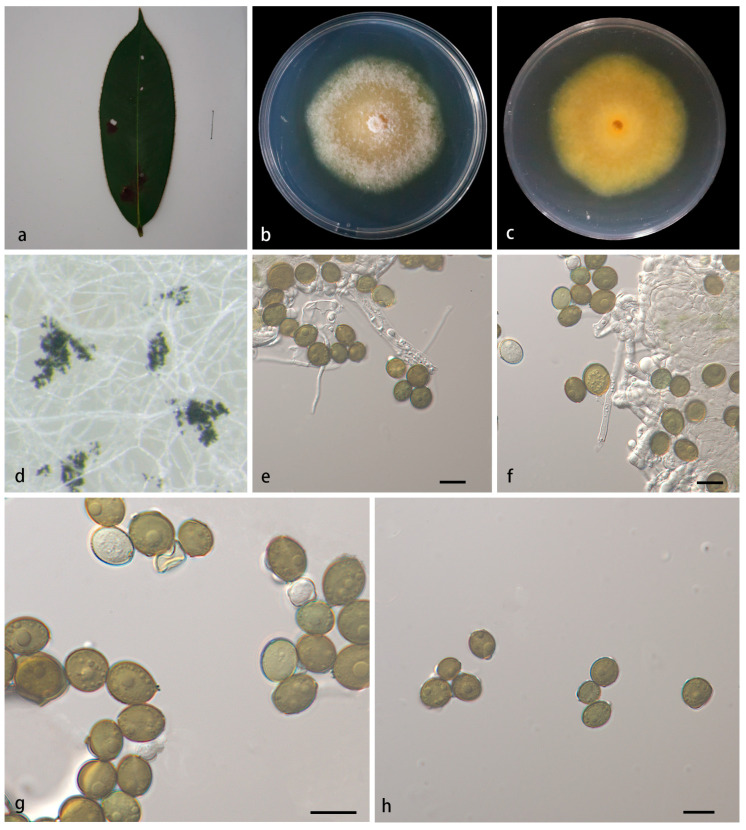
*Apiospora adinandrae* (HMAS 352657, holotype) (**a**) diseased leaf of *Adinandra glischroloma*, (**b**,**c**) colonies after 14 days on PDA ((**b**), obverse; (**c**), reverse), (**d**) colony overview, (**e**,**f**) conidia with conidiogenous cells, (**g**,**h**) conidia. Scale bars: 10 μm (**e**–**h**).

#### 3.2.2. *Apiospora bawanglingensis* X.Y. Liu, Z.X. Zhang and X.G. Zhang, sp. nov. (Figure 3)

MycoBank—No: 850661;Etymology—The specific epithet “*bawanglingensis*” refers to the Bawangling National Forest Park, where the type was collected;Type—Bawangling National Forest Park, Hainan Province, China, on diseased leaves of *Indocalamus longiauritus*, 19 May 2021, X.Y. Liu (HMAS 352654, holotype), ex-holotype living culture SAUCC BW0444;Description—On PDA, hyphae 3.0–3.5 μm in diameter, branched, hyaline and septate. Sexual morph: Undetermined. Asexual morph: Conidiophore reduced to conidiogenous cells, aggregated in clusters on hyphae. Conidiogenous cells dark green, becoming brown, polyblastic, cylindrical, septate, verrucose, flexuous, 5.0–10.0 × 2.0–3.0 μm. Conidia smooth, rounded to ovoid, globose to subglobose, green to dark brown, 6.4–7.7 × 5.3–7.1 μm, mean ± SD = 7.1 ± 0.4 × 6.1 ± 0.5 μm, n = 30. See Figure 3;Culture characteristics—PDA, colonies concentrically spreading, fluffy, with abundant aerial sparse mycelium, white to cream. In reverse, white, becoming tawny from the center. After seven days of incubation at 25 °C, the colony diameter reached 73.7–82.5 mm and the growth rate was 10.5–11.7 mm/day;Additional specimen examined—China, Hainan Province: Bawangling National Forest Park, on diseased leaves of *Indocalamus longiauritus*, 19 May 2021, X.Y. Liu, HSAUP BW04441, living culture SAUCC BW04441;Notes—Phylogenetic analyses based on ITS-LSU-*TEF1α*-*TUB2* rDNA sequences showed that *Apiospora bawanglingensis* sp. nov. formed an independent clade which is closely related to *A. piptatheri* (CBS 145149). The base-pair comparison of ITS, LSU and *TEF1α* sequences, respectively, showed 2.74%, 0.72% and 8.63% differences between *A. bawanglingensis* (SAUCC BW0444) and *A. piptatheri* (CBS 145149);Morphologically, *A. bawanglingensis* differs from *A. piptatheri* in conidiogenous cells and conidia. The conidiogenous cells of *A. piptatheri* discrete, sometimes branched, measure 6.0–27.0 × 2.0–5.0 µm. Compared to *A. piptatheri*, *A. bawanglingensis* has brown, polyblastic, cylindrical, septate, shorter conidiogenous cells (5.0–10.0 × 2.0–3.0 μm). The conidia of *A. piptatheri* (6.0–8.0 × 3.0–5.0 μm) and *A. bawanglingensis* (6.4–7.7 × 5.3–7.1 μm) are similar in shape, but the conidia of *A. piptatheri* with a thin hyaline germ-slit and the conidia of *A. bawanglingensis* are not observed [4]. For details, see Table 3.

**Figure 3 jof-10-00074-f003:**
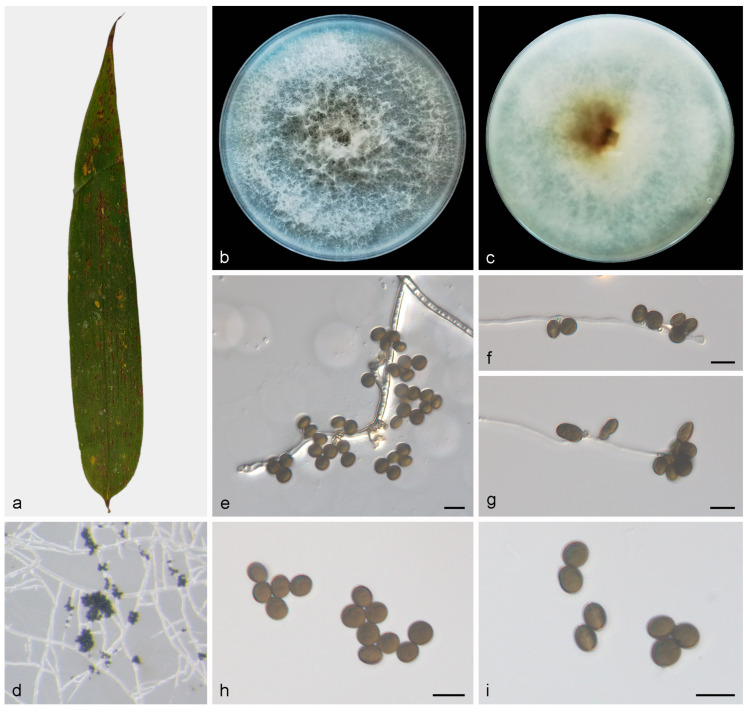
*Apiospora bawanglingensis* (HMAS 352654, holotype) (**a**) diseased leaf of *Indocalamus longiauritus*, (**b**,**c**) colonies after 14 days on PDA ((**b**), obverse; (**c**), reverse), (**d**) colony overview, (**e**–**g**) conidia with conidiogenous cells, (**h**,**i**) conidia. Scale bars: 10 μm (**e**–**i**).

#### 3.2.3. *Apiospora machili* X.Y. Liu, Z.X. Zhang and X.G. Zhang, sp. nov. (Figure 4)

MycoBank—No: 850665;Etymology—The epithet *machili* pertains to the generic name of the host plant *Machilus nanmu*;Type—Wuyishan National Forest Park, Fujian Province, China, on diseased leaves of *Machilus nanmu*, 15 October 2022, X.Y. Liu (HMAS 352656, holotype), ex-holotype living culture SAUCC 1175A–4;Description—On PDA, hyphae 2.5–3.5 μm in diameter, branched, hyaline and septate. Sexual morph: Undetermined. Asexual morph: Conidiophore reduced to conidiogenous cells, aggregated in clusters on hyphae. Conidiogenous cells pale green, becoming brown, polyblastic, cylindrical, septate, verrucose, flexuous, 6.0–8.0 × 2.5–4.0 μm. Conidia smooth, green to dark brown, 7.1–9.5 × 5.6–8.8 μm, mean ± SD = 8.5 ± 0.6 × 7.7 ± 0.7 μm. In face view, rounded to ovoid, globose to subglobose; in side view, lenticular, with a pale equatorial slit. Sexual morph: Undetermined. See Figure 4;Culture characteristics—PDA, colonies concentrically spreading with undulate margin, wooly aerial mycelium, flat, ivory. In reverse, the whole is ivory, with a slight yellow in the center. After seven days of incubation at 25 °C, the colony diameter reached 69.7–78.8 mm and the growth rate was 9.9–11.2 mm/day;Additional specimen examined—Wuyishan National Forest Park, Fujian Province, China, on diseased leaves of *Machilus nanmu*, 15 October 2022, X.Y. Liu, HSAUP 1175, living culture SAUCC 1175;Notes—Phylogenetic analyses based on ITS-LSU-*TEF1α*-*TUB2* rDNA sequences showed that *Apiospora machili* sp. nov. formed an independent clade which is closely related to *A. setariae* (CFCC 54041) and *A. jiangxiensis* (LC4577). The base-pair comparison of ITS, LSU, *TEF1α* and *TUB2*, respectively, showed 3.40%, 0%, 0% and 0% differences between *A. machili* (SAUCC 1175A-4) and *A. setariae* (CFCC 54041), and showed 2.2%, 0.24%, 4.35% and 4.54% differences between *A. machili* and *A. jiangxiensis* (LC4577);Morphologically, *A. machili* differs from *A. setariae* and *A. jiangxiensis* in conidiophore, conidiogenous cells and conidia. *A. setariae* has erect conidiophore and longer conidiogenous cells (8.0–55.0 × 1.0–3.5 μm), whereas *A. machili* conidiophore reduced to conidiogenous cells (6.0–8.0 × 2.5–4.0 μm) and the conidiogenous cells polyblastic, cylindrical, flexuous. *A. machili* differs from *A. jiangxiensis* by its cylindrical conidiogenous cells (6.0–8.0 × 2.5–4.0 μm), while *A. jiangxiensis* has conidiogenous cells that are clearly ampulliform (apical neck 2.5–6.0 µm, basal part 3.0–9.0 µm). *A. machili*, *A. setariae* and *A. jiangxiensis* have morphologically similar conidia (7.1–9.5 × 5.6–8.8 μm vs. 7.5–10.5 μm vs. 7.5–10.0 × 4.5–7.0 μm) [4]. For details, see Table 3.

**Figure 4 jof-10-00074-f004:**
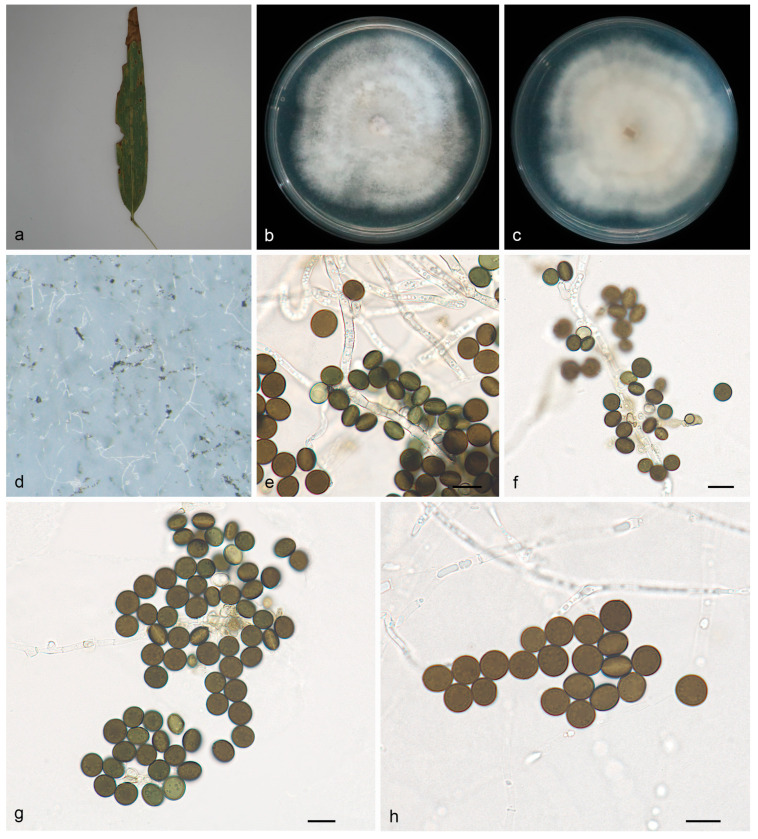
*Apiospora machili* (HMAS 352656, holotype) (**a**) diseased leaf of *Machilus nanmu*, (**b**,**c**) colonies after 14 days on PDA ((**b**), obverse; (**c**), reverse), (**d**) colony overview, (**e**,**f**) conidia with conidiogenous cells, (**g**,**h**) conidia. Scale bars: 10 μm (**e**–**h**).

#### 3.2.4. *Apiospora piptatheri* Pintos and P. Alvarado, MycoKeys 49: 40. (2019) (Figure 5)

Description—On PDA, hyphae 2.5–4.0 μm in diameter, branched, hyaline and septate. Asexual morph: Conidiophore reduced to conidiogenous cells, aggregated in clusters on hyphae. Conidiogenous cells pale green, becoming brown, polyblastic, cylindrical, septate, verrucose, flexuous, 10.0–15.0 × 3.0–5.0 μm. Conidia smooth, rounded to ovoid, globose to subglobose, green to dark brown, 6.2–7.6 × 4.9–7.2 μm, mean ± SD = 6.9 ± 0.4 × 6.1 ± 0.6 μm. Sexual morph: Undetermined. See Figure 5;Culture characteristics—PDA, colonies concentrically spreading with irregular margin, abundant fluffy aerial mycelium which can fill a plate, mycelium white to cream. In reverse, the central part is brown to yellow, and gradually lightened outward, from yellow to white. After seven days of incubation at 25 °C, the colony diameter reached 73.7–82.5 mm and the growth rate was 10.5–11.7 mm/day;Additional specimen examined—Bawangling National Forest Park, Hainan Province, China, on diseased leaves of *Indocalamus longiauritus*, 19 May 2021, X.Y. Liu, HSAUP BW04551, living culture SAUCC BW04551;Notes—Phylogenetic analyses based on ITS-LSU-*TEF1α* rDNA sequences showed that *Apiospora piptatheri* (SAUCC BW0455) formed an independent clade which is closely related to *A. piptatheri* (CBS 145149). The base-pair comparison of ITS, LSU and *TEF1α* showed 0.36%, 0.96% and 56.2% differences between SAUCC BW0455 and CBS 145149, respectively;Morphologically, strain SAUCC BW0455 and *A. piptatheri* (CBS 145149) have similar characteristics. The isolates (SAUCC BW0455) are similar to the type strain of *A. piptatheri* (CBS 145149) in having smooth, subglobose, brown conidia (6.2–7.6 × 4.9–7.2 μm vs. 6.0–8.0 × 3.0–5.0 μm); however, no germ-slit were observed in SAUCC BW0455 [4]. For details, see Table 3.

**Figure 5 jof-10-00074-f005:**
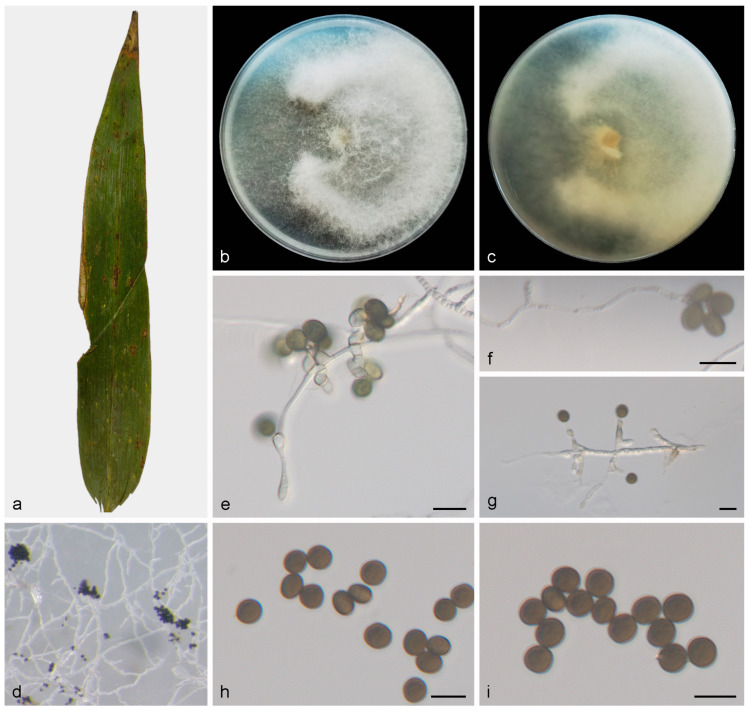
*Apiospora indocalami* (HMAS 352655, holotype) (**a**) diseased leaf of *Indocalamus longiauritus* (**b**,**c**) colonies after 14 days on PDA ((**b**), obverse; (**c**), reverse) (**d**) colony overview (**e**–**g**) conidia with conidiogenous cells (**h**,**i**) conidia. Scale bars: 10 μm (**e**–**i**).

**Table 3 jof-10-00074-t003:** The asexual morphological characters of some *Apiospora* species.

Strain	Host	Country	Conidiogenous Cells	Conidia in Surface View	**Size (μm)**	**References**
*Apiospora descalsii*	*Ampelodesmos mauritanicus*	Spain	solitary on hyphae, ampulliform, hyaline	brown, smooth, guttulate, globose, ellipsoid	(5.0–) 7.0 (–8.0)	[11]
*A. esporlensis*	*Poaceae*	Spain	polyblastic, aggregated, smooth, hyaline, ampuliform, cylindrical or lageniform	brown, smooth, globose, pale equatorial slit	(8.0–) 9.0–12.0 (–13.0)	[11]
*A. iberica*	*Poaceae*	Portugal	aggregated or solitary, ampulliform or cylindrical	brown, smooth, globose to ellipsoid	(9.0–) 10.0 (–12.0)	[11]
*A. italica*	*Poaceae*	Italy	ampulliform, cylindrical or doliiform, hyaline to brown	brown, smooth, globose	(3.0–) 4.0–7.0 (–9.0) × (1.5–) 2.0–3.0 (–5.0)	[11]
*A. piptatheri*	*Poaceae*	Spain	basauxic, polyblastic, sympodial, cylindrical, discrete, branched, smooth	globose, ellipsoidal, brown, with a thin hyaline germ-slit	6.0–8.0 × 3.0–5.0	[11]
*A. fermenti*	*seaweed*	Korea	-	Globose to elongate, ellipsoid	7.8–8.8 × 7.2–8.6	[13]
*A. hainanensis*	*Poaceae*	China	globose, erect, blastic, branched, aggregated, hyaline, pale brown, smooth	globose to lenticular, longitudinal germ-slit, ellipsoidal, brown, smooth to finely roughened	5.5–8.5 × 5.0–7.5	[14]
*A. dongyingensis*	*Poaceae*	China	globose, erect, blastic, aggregate, hyaline, pale brown, smooth, branched	globose, lenticular, longitudinal germ-slit, elongated, brown, smooth to finely roughened	8.0–16.5 × 5.5–9.0	[14]
*A. acutiapica*	*Poaceae*	China	cylindrical to ampulliform, pale brown	hyaline apex, smooth, dark equatorial slit	-	[15]
*A. chiangraiense*	*Poaceae*	Thailand	smooth, monoblastic or polyblastic, aggregated, light brown, cylindrical	aseptate, pale brown to dark brown	6.5–8.0 × 6.0–8.0	[15]
*A. thailandica*	*Poaceae*	Thailand	basauxic, polyblastic, smooth, sympodial, cylindrical, discrete, sometimes branched	globose, occasionally elongated, dark brown, smooth, with a truncate basal scar	5.0–9.0 × 5.0–8.0	[16]
*A. yunnana*	*Poaceae*	China	basauxic, cylindrical, discrete, smooth	lenticular, obovoid, dark brown, smooth, with a truncate basal scar	17.5–26.5 × 15.5–25.0	[16]
*A. camelliae-sinensis*	*Camellia sinensis*	China	erect, aggregated, doliiform to ampulliform, pale brown, smooth	brown, smooth, globose	9.0–13.5 × 7.0–12.0	[17]
*A. dichotomanthi*	*Dichotomanthus tristaniaecarpa*	China	erect, aggregated on hyphae, doliiform to clavate or lageniform, hyaline, pale brown, smooth	brown, smooth to finely roughened, globose, with a longitudinal germ-slit	9.0–15.0 × 6.0–12.0	[17]
*A. guizhouensis*	*Air*	China	erect, aggregated, pale brown, smooth, subglobose, ampulliform or doliiform	dark brown, smooth to finely roughened, globose, germ-slit	5.0–7.5 × 4.0–7.0	[17]
*A. jiangxiensis*	*Maesa* sp.	China	erect, scattered or aggregated, hyaline, smooth, ampulliform	brown, smooth to finely roughened, granular, globose	7.5–10.0	[17]
*A. neobambusae*	*Poaceae*	China	erect, aggregated, hyaline, smooth, doliiform to ampulliform, or lageniform	olivaceous, smooth to finely roughened, subglobose	11.5–15.5 × 7.0–14.0	[17]
*A. obovata*	*Lithocarpus* sp.	China	erect, aggregated, pale brown, smooth, subcylindrical or clavate	dark brown, roughened, globose	11.0–16.5	[17]
*A. pseudoparenchymatica*	*Poaceae*	China	pale yellow, finely roughened, subcylindrical to doliiform	dark brown, smooth, finely guttulate, globose	13.5–27.0 × 12.0–23.5	[17]
*A. subrosea*	*Poaceae*	China	pale brown, smooth, doliiform to subcylindrical	brown, smooth, subglobose or ellipsoidal	12.0–17.5 × 9.0–16.0	[17]
*A. pseudorasikravindrae*	*Poaceae*	China	holoblastic, ampulliform, cylindrical or doliiform, olivaceous	globose	5.0–10 × 5.5–11.0	[20]
*A. agari*	*Agarum cribrosum*	Korea	clusters or solitary, hyaline becoming pale green, cylindrical, ampulliform	smooth to granular, globose to subglobose	(8.5–) 9.0–10.5 × (7.0–) 7.5–8.5 (−9.0)	[37]
*A. arctoscopi*	*egg masses of Arctoscopus japonicus*	Korea	clusters or solitary, hyaline, cylindrical, ampulliform	brown, smooth to granular, globose to elongate, ellipsoid	(9.5–) 10.0–12.0 (−13) × (7.5–) 8.0–11.0 (−12.0)	[37]
*A. koreana*	*egg masses of Arctoscopus japonicus*	Korea	aggregated in clusters on hyphae, hyaline, cylindrical	brown, smooth to granular, globose to ellipsoid	(7.5–) 8.0–10 (−11) × (5.5–) 6.5–9.5 (−10)	[37]
*A. marina*	*seaweed*	Korea	aggregated or solitary, hyaline, erect, ampulliform	globose to elongate ellipsoid, brown, smooth to granular	(9.5–) 10.0–12.0 (−13.0) × (7.5–) 8.0–10.0	[37]
*A. pusillisperma*	*Seaweed*	Korea	aggregated, hyaline, cylindrical	brown, smooth to granular, globose	4.0–6.0 (−6.5) × (3.0–) 3.5–5.0 (−5.5)	[37]
*A. sargassi*	*Sargassum fulvellum*	Korea	aggregated or solitary, hyaline, basauxic, polyblastic, sympodial, erect, cylindrical	brown, smooth to granular, globose	(8.5–) 9.5–11.0 (−11.5) × (8.0–) 8.5–10 (−11)	[37]
*A. aquatica*	*decaying wood*	China	erect, aggregated in clusters, smooth, doliform to ampulliform	globose to subglobose, smooth, olivaceous to brown	9.0–11.0 × 8.0–10.0	[38]
*A. bambusicola*	*Poaceae*	Thailand	polyblastic, terminal, cylindrical, smooth, aggregated, light brown	solitary, oval or irregularly round, brown, guttulate, granular	6.0–8.0 × 6.0–7.8	[39]
*A. biserialis*	*Poaceae*	China	integrated, pale brown, doliiform to ampulliform, or lageniform	brown, smooth	7.0–9.0	[40]
*A. cyclobalanopsidis*	*Cyclobalanopsis glauca*	China	aggregated, pale brown, ampulliform or cylindrical	brown, smooth, globose to ellipsoid	8.0–12.0	[40]
*A. septata*	*Poaceae*	China	solitary, integrated, branched, ampulliform, cylindrical, brown	brown, smooth, guttulate, globose	8.0–11.0 (−13.0)	[40]
*A. chromolaenae*	*Poaceae*	Thailand	basauxic, broadly filiform to ampulliform, aggregated, hyaline, smooth, elongated,	irregular arrangement, pale brown, smooth, globose	4.0–6.0 × 4.5–6.5	[41]
*A. cordylines*	*Cordyline fruticosa*	China	erect, aggregated into clusters, hyaline, smooth, lageniform	olivaceous, smooth to finely roughened, subglobose, ellipsoid	15.0–19.0 × 12.5–18.5	[42]
*A. gaoyouensis*	*Poaceae*	China	aggregated, smooth, short and wide	brown, smooth, granular, globose to elongate ellipsoid	5.0–8.0	[43]
*A. qinlingensis*	*Poaceae*	China	aggregated in clusters on hyphae, smooth, short	brown, smooth, granular, globose to suborbicular	5.0–8.0	[43]
*A. guiyangensis*	*Poaceae*	China	solitary, integrated, branched, ampulliform, cylindrical, hyaline	brown, smooth, guttulate, globose to ellipsoid	10.0–13.0 × 7.0–10.5	[44]
*A. hydei*	*Trachycarpus fortune*	China	aggregated, brown, smooth, subcylindrical to doliiform to lageniform	brown, finely roughened, globose	(15.0–) 17.0–19.0 (–22.0)	[45]
*A. kogelbergensis*	*Poaceae*	south Africa	aggregated, pale brown, smooth, doliiform to subcylindrical	globose to ellipsoid	9.0–10.0 × 7.0–8.0	[45]
*A. malaysiana*	*Macaranga hullettii*	Malaysia	aggregated, hyaline, pale brown, smooth, doliiform to clavate to ampulliform	brown, smooth, globose	5.0–6.0	[45]
*A. ovata*	*Poaceae*	China	pale brown, smooth, aggregated, ampulliform	broadly ellipsoid, medium brown, finely roughened	18.0–20.0	[45]
*A. pseudosinensis*	*Poaceae*	Netherlands	doliiform or subcylindrical, pale brown, smooth	brown, smooth, ellipsoid,	8.0–10.0 × 7.0–10.0	[45]
*A. pseudospegazzinii*	*Macaranga hullettii*	Malaysia	aggregated, brown, smooth, ampulliform with elongated neck	brown, guttulate, roughened, globose	8.0–9.0	[45]
*A. vietnamensis*	*Macaranga hullettii*	Malaysia	aggregated, pale brown, smooth, doliiform to clavate, ampulliform	aggregated, brown and globose	5.0–6.0	[45]
*A. xenocordella*	*soil*	Austria	aggregated, brown, verruculose, globose to clavate to doliiform,	brown, smooth, guttulate, globose to ellipsoid	(7.0–) 9.0–10.0 (–11.0)	[45]
*A. hyphopodii*	*Poaceae*	China	basauxic, cylindrical, discrete, with verrucose wall	globose, dark brown, smooth, truncate scar, with a longitudinal, germ-slit	4.0–6.0 × 2.0–3.5	[46]
*A. locuta-pollinis*	*honey bee colonies*	China	pale brown, smooth, subglobose to ampulliform to doliiform	pale brown with hyaline equatorial rim, smooth, globose	5.5–9.0 × 4.5–8.0	[47]
*A. marianiae*	*Poaceae*	Spain	monoblastic, integrated, terminal, intercalary, cylindrical	brown, solitary	(11.0–) 12.1–13.5 (–18.0)	[48]
*A. montagnei*	*Arundo micrantha*	Spain	doliiform to lageniform or ampulliform, hyaline	ellipsoidal to ovoid, smooth to finely roughened, with an equatorial germ-slit of paler pigment	(9.0–) 10.3–11.3 (–12.0)	[48]
*A. minutispora*	*soil*	Korea	erect, ellipsoid to ovoid, hyaline, pale brown to umber in color, and smooth	brown, finely roughened, ellipsoidal to ovoid, thick, solitary or aggregated, irregular dot-like structures	5.7–8.2 × 4.6–7.0	[49]
*A. mori*	*Morus australis*	China	pale yellow, smooth or finely roughened, subcylindrical to doliiform	globose, dark brown, smooth, with a basal scar, occasionally with germ-slit	4.5–5.5 × 4.0–5.0	[50]
*A. neogarethjonesii*	*Poaceae*	China	basauxic, cylindrical, discrete, smooth-walled	globose, dark brown, smooth, with a truncate basal scar	20–35 × 15–30	[51]
*A. paraphaeosperma*	*Poaceae*	Thailand	basauxic, aggregated, hyaline, smooth, elongated, conical	brown, smooth, granular, globose to ellipsoid	10.0–19.0	[52]
*A. phyllostachydis*	*Poaceae*	China	holoblastic, monoblastic, cylindrical, hyaline to pale brown, smooth, thin-walled	globose, irregular, pale brown, guttulate, olive to dark brown, with a germ-slit, smooth	25.0–35.0 × 20.0–25.0	[53]
*A. rasikravindrae*	*soil*	Norway	mononematous, hyaline, straight or flexuous, thin-walled, unbranched, septate, smooth	lenticular, ovoid	10.0–15.0 × 6.0–10.5	[54]
*A. stipae*	*Poaceae*	Spain	aggregated, pale brown, smooth, ampulliform	red-brown, thick-walled, smooth, eguttulate, with lateral germ-slit, often with a pronounced hilum	6.5–10.5 × 6.0–9.0	[54]
*A. sasae*	*Poaceae*	Netherlands	discrete, subcylindrical, subhyaline, proliferating sympodially, smooth to finely verruculose, holoblastic	numerous, aseptate, subglobose, thick-walled, smooth, with a lateral hyaline germ-slit	(16.0–) 17.0–18.0 (–20.0) × (15.0–) 16.0–17.0 (–19.0)	[55]
*A. setariae*	*Poaceae*	China	erect, hyaline to pale brown, smooth	globose, oval or irregular, brown, guttulate, with a longitudinal germ-slit	7.5–10.5	[56]
*A. setostroma*	*Poaceae*	China	micronematous, holoblastic, monoblastic, hyaline, cylindrical, flexible, discrete, aseptate, smooth	acrogenous, dark brown, obovoid, septate, smooth, multi-guttulate, with a scar	18–20 × 15–19	[57]
*A. sorghi*	*Poaceae*	Brazil	aggregated, hyaline, cylindrical to subcylindrical	brown, smooth, globose, subglobose, with a longitudinal germ-slit	6.0–8.0 × 6.0–10.0	[58]
** *A. adinandrae* **	*Adinandra glischroloma*	China	pale green, cylindrical, septate, flexuous	smooth, rounded to ovoid, green to dark brown	7.5–12.4 × 6.3–10.5	**This study.**
** *A. bawanglingensis* **	*Poaceae*	China	pale green, polyblastic, cylindrical, septate, verrucose, flexuous	smooth, green to dark brown, rounded to ovoid, globose to subglobose,	6.4–7.7 × 5.3–7.1	**This study.**
** *A. piptatheri* **	*Poaceae*	China	pale green becoming brown, polyblastic, cylindrical, septate, verrucose, flexuous	smooth, rounded to ovoid, globose, green to dark brown	6.2–7.6 × 4.9–7.2	**This study.**
** *A. machili* **	*Machilus nanmu*	China	pale green, becoming brown, polyblastic, cylindrical, septate, verrucose, flexuous,	smooth, rounded to ovoid, globose, green to dark brown,	7.1–9.5 × 5.6–8.8	**This study.**

Notes: the species information described in this study is marked in bold.

## 4. Discussion

In this study, three new species, viz., *Apiospora adinandrae* sp. nov., *A. bawanglingensis* sp. nov. and *A. machili* sp. nov., are introduced and described based on their morphological characters and phylogenetic status. *A. bawanglingensis* was collected from diseased leaves of *Indocalamus longiauritus* in Bawangling National Forest Park, Hainan Province, China. *A. adinandrae* and *A. machili* were collected from Wuyi Mountain, Fujian Province, China; the host of the former was *Adinandra glischroloma* and the latter was *Machilus nanmu*.

Phylogenetically, most studies have used phylogenetic analyses of ITS, LSU, *TUB2* and *TEF1α* sequence data to identity *Apiospora* species. Since the ITS and LSU gene regions are relatively conserved, the *TUB2* and *TEF1α* gene regions have played an important role in the species identification of *Apiospora* [59]. In the present study, we constructed phylogenetic trees for each gene of ITS, LSU, *TEF1α* and *TUB2* as Appendix A to show the congruence (Appendix A). The phylogenetic analyses showed *Apiospora adinandrae* sp. nov., *A. bawanglingensis* sp. nov. and *A. machili* sp. nov. all as a monophyletic clade (Figure 1), and the comparisons of both ITS and LSU showed small differences. In the *A. adinandrae* clade, the *A. adinandrae* is distinguished from *A. aurea* by 6/590, 4/831, 29/426 and 16/442 nucleotides; from *A. hydei* by 7/589, 3/831, 29/427 and 21/442; and from *A. cordylines* by 6/574, 0/831, 28/433 and 22/442 in the ITS, LSU, *TEF1α* and *TUB2* sequences, respectively. In the *A. bawanglingensis* clade, *A. bawanglingensis* is distinguished from *A. piptatheri* by 16/583, 6/829, 39/452 and 0/449 characters, respectively. In the *A. machili* clade, the *A. machili* is distinguished from *A. setariae* by 20/588, 0/830, 0/437 and 0/441 characters and from *A. jiangxiensis* by 13/588, 2/830, 19/437 and 20/441 in the ITS, LSU, *TEF1α* and *TUB2* sequences, respectively.

The asexual morphology of the species in this study is consistent with the basic characteristics of the genus. Morphologically, the biological relationship between *Apiospora* and *Arthrinium* has been controversial due to their similar morphological characteristics in having basauxic conidiogenesis [57]. Although some morphological features of *Arthrinium* species are difficult to observe in *Apiospora*, such as the conidiophores having black thick septa, and it is still difficult to define the boundary between *Apiospora* and *Arthrinium* by the asexual form alone [4,58]. It has been demonstrated that ascogenous *Apiospora* can reproduce the mycelial asexual state on artificial media such as PDA and MEA [57,60]. Despite this cultural and molecular evidence of asexual to sexual transition, the sexual morphology of *Apiospora* is still rarely reported [61].

Ecologically, *Apiospora* is widely distributed in subtropical, tropical, temperate and even cold regions, including Africa, America, Asia, Australia, and Europe, according to reported data [13,15]. As endophytes, plant pathogens and humus, *Apiospora* is ubiquitous in a wide range of terrestrial environments, such as soil, atmosphere and even marine substrates, but its main hosts are still plants, especially *Poaceae* [60]. Of all the *Apiospora* species that have been reported, more than 60% of the hosts are *Poaceae*, of which about half are from bamboos [59]. The new species *A. bawanglingensis* in this paper was also collected from *Indocalamus longiauritus*, which further enriches the diversity of bamboo fungi. Based on the existing statistical data of the USDA fungal database (https://nt.ars-grin.gov/fungaldatabases/, accessed on 10 November 2023.) and the collation of related literature published later on in the genus *Apiospora*, only 14 records of the genus *Apiospora* were isolated from woody plants (trees, shrubs, small shrubs), accounting for less than 10% of the total records, and about half of these hosts are *Arecaceae*. The strains *A. adinandrae* and *A. machili* in this study were isolated from *Adinandra glischroloma* and *Machilus nanmu*, respectively, which is the first time that *Apiospora* has been isolated from non-*Poaceae* hosts in China after Lu, B. (2000) [62].

Based on morphological and phylogenetic analysis, our study identified three new species in which only asexual morphology was found. *Apiospora adinandrae* sp. nov., *A. bawanglingensis* sp. nov. and *A. machili* sp. nov. are morphologically similar to their sister taxa, with significant differences in sequence. *A. bawanglingensis* was collected from *Indocalamus longiauritus*. Because most fungi on bamboo were not pathogens and there was no obvious plaque on the host, we believe for the time being that *A. bawanglingensis* is not a pathogen, and its pathogenicity still needs further study. *A. adinandrae* and *A. machili* were isolated from *Adinandra glischroloma* and *Machilus nanmu*, respectively. Based on the existing statistical data of the USDA fungal database, this is the first time that *Apiospora* has been found on *Adinandra glischroloma* and *Machilus nanmu*, enriching the host diversity of *Apiospora*.

## Figures and Tables

**Figure 1 jof-10-00074-f001:**
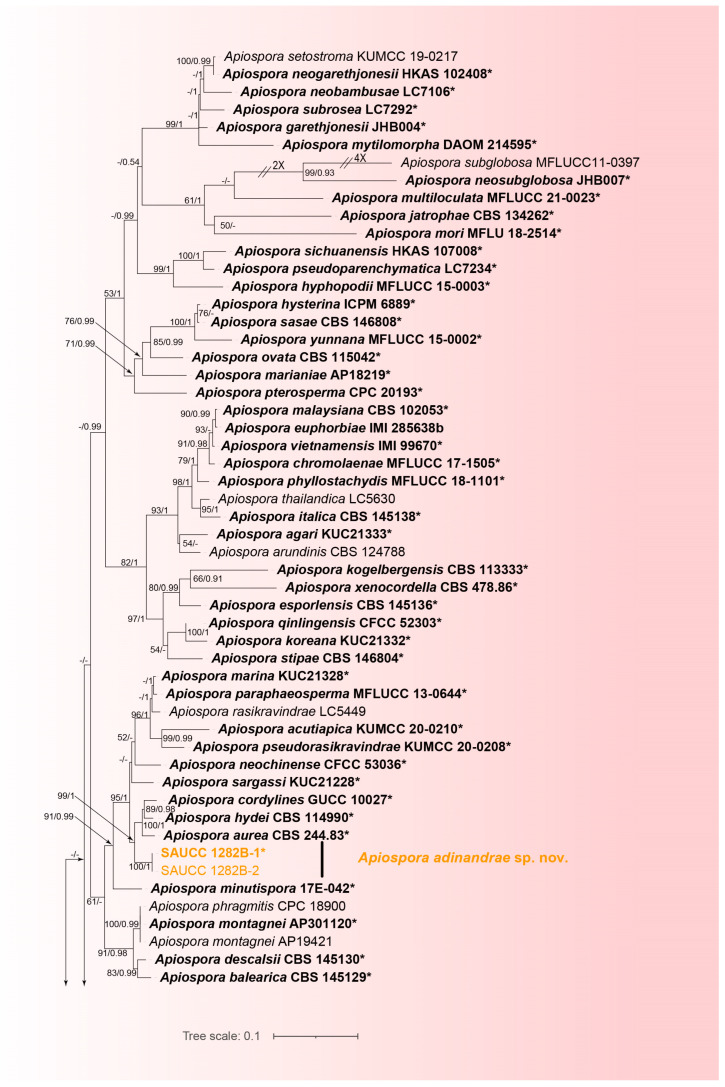
A maximum likelihood tree showing the phylogenetic relationships of *Apiospora* inferred from the ITS, LSU, *TEF1α* and *TUB2* sequences, and the roots on *Arthrinium caricicola* (CBS 145127). The Bayesian inference posterior probability (left, BIPP ≥ 0.90) and the maximum likelihood bootstrap value (right, MLBV ≥ 50%) are shown as BIPP/ML above the nodes. Strains marked with a star “*” and bolded represented are ex-types or ex-holotypes. Strains from the present study are in orange. The scale in the bottom middle indicates 0.1 substitutions per site. In order to make the layout of the evolutionary tree beautiful, some branches are shortened by two diagonal lines (“/”) with the number of times.

**Table 1 jof-10-00074-t001:** Strains associated with host and region in this study.

Strain	Host	Region
BW0444	*Indocalamus longiauritus*	Hainan Province
BW04441	*Indocalamus longiauritus*	Hainan Province
BW0455	*Indocalamus longiauritus*	Hainan Province
BW04551	*Indocalamus longiauritus*	Hainan Province
XG01282B-1	*Adinandra glischroloma*	Fujian Province
XG01282B-2	*Adinandra glischroloma*	Fujian Province
XG01175A-4	*Machilus nanmu*	Fujian Province
XG01175	*Machilus nanmu*	Fujian Province

## Data Availability

The sequences from the present study were submitted to the NCBI database (https://www.ncbi.nlm.nih.gov/, accessed on 20 November 2023) and the accession numbers were listed in Appendix A.

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
