# Peer review of "Three New Species of Apiospora (Amphisphaeriales, Apiosporaceae) on Indocalamus longiauritus, Adinandra glischroloma and Machilus nanmu from Hainan and Fujian, China"

_jof, 2024, doi:10.3390/jof10010074_

Round 1

Reviewer 1 Report

Comments and Suggestions for Authors

3.2.1, 2, 3 species were written in italic.

Reduce the Table 3, leaving only related species.

The format of Table 3 is the same as Table 1. 

Author Response

Dear Reviewer,

Thank you for your valuable suggestion. In response to these questions, I answer as follows:

  1. 2.1, 2, 3 species were written in italic.

I agree with the above suggestions.

  1. Reduce the Table 3, leaving only related species.

Thank you for your advice, but after consideration I decided to keep Table 3. The host, geographical location, and conidial morphology and size of species in Apiospora are discussed in this paper, so I have compiled this table to visualize the differences between them. If information about other species is removed, the species described in this paper will not be well compared with known species

  1. The The format of Table 3 is the same as Table 1.

I accepted the suggestions.

Best wishes,

Xinye Liu

Reviewer 2 Report

Comments and Suggestions for Authors

Congratulations for your work. You isolated several strains of endophyte fungi and analyzed morphologically and also with DNA sequences. However as a reviewer I need to be critic, the limited number of samples per species limits the strenght of your results. You should contrast the congruence between single locus trees and include single locus tree as supplementary figures.

 The introduction could be improved and the discussion should be.

And last but not least I am not convinced of the proposal of A. indocalami as new species.

  If you avoid to propose A. indocalami as new species and you show the single locus trees my major concerns will be out. The rest are almost aestetic changes proposed to the authors to improve the quality of the manuscript.

FInd my detailed comments in the attached file

Author Response

Dear Reviewer,

Thank you so much for your valuable suggestions. I accepted your suggestions and revised the content of the article according to your suggestions, and your modification of the format and grammar is of great help to me.

In addition to the errors and omissions in the format, I have greatly modified the relevant content of Apiospora indocalami sp. nov.. At your prompt, I rechecked the sequence Apiospora indocalami and A. piptatheri (CBS 145149) used for comparison and found that the sequence I used in notes was not CBS 145149, which I corrected. The base comparison in notes was changed to a percentage as you suggested, and the specific number of nucleotide was shown in the discussion. At the same time, I added phylogenetic analysis to the discussion.

I constructed single-gene phylogenetic trees for each of the four gene loci and included them in the supplementary materials. The strains and Genbank code used in this paper are arranged in supplementary material Table S1.

After re-reading the published literature, I found that it is permitted to attach direct links to web pages in the article, so I think the URL in the article can be retained.

When we constructed the phylogenetic tree, we made reference to the construction of evolutionary trees in other literature of this genus, and finally used all species in this genus, and most of them chose the sequence of type species. We believed that adding sequences would cause unnecessary duplication, because the sequences of different strains of the same species were basically the same.

I have deleted part of the content in the discussion, and most of the discussion on the difference between Apiospora and Arthrinium and the hosts has been retained. In order to avoid excessive repetition, these two parts are briefly mentioned in the introduction, and my personal opinions and findings are put in the discussion.

Thank you so much again, and your suggestions have made me rethink my work more deeply and have greatly helped me improve my logical thinking and text details in writing. In the future, I will take a more rigorous attitude towards my work and try to improve my critical thinking ability. I have benefited a lot from your suggestions, and I am very appreciated for your careful review and reply, which makes me learn a lot.

Your prompt attention to this letter will be highly appreciated.

Best wishes,

Xinye Liu

Reviewer 3 Report

Comments and Suggestions for Authors

Dear authors of the manuscript "Four New Species of Apiospora (Amphisphaeriales, Apiosporaceae) on Indocalamus longiauritus, Adinandra glischroloma and Machilus nanmu from Hainan and Fujian, China”, after evaluating your manuscript, I recommend the following adjustments and major revision:

·    Include more Apiospora piptatheri sequences in phylogenetic analyzes to ensure that A. indocalami sp. nov. will not be positioned within the clade of A. piptatheri. Based on the phylogeny and also on the fact that the only morphological difference you pointed out between these species was tenths of a micrometer in conidial measurements, I consider it necessary to provide more robust evidence for the separation of these species. Then try this by including more A. piptatheri sequences in the tree and point out more morphological differences between the species in the Notes section after the description.

·  In the descriptions of the four species, try to characterize the conidiogenic cells in more detail. In the notes after the descriptions provide further morphological differences between the described species and their respective sister taxa.

·      Improve the legends of the five figures (See examples in taxonomy articles).

·         Page 1, line 2: Apiospora should be written in italics.

·       Page 1, lines 15-16 and where else the genes were mentioned in the text, tables, and figure legends: Genes abbreviations TEF1-a and TUB2 should be italicized. This is not necessary for ITS and LSU as they are not protein coding regions.

·   Page 1, line 17: Replace "Combined with morphological and multi-site phylogenetic analysis" with "Based on morphological and molecular phylogenetic analyses".

·    Page 1, line 20: Do not use words contained in the title as keywords (Apiospora).

·         Page 1, line 24: It is Saccardo, not Sacardo.

·  Page 1, lines 25-26: Review the sentence, something in it is meaningless.

·   Page 2, lines 57-58: Replace "To clarify taxonomic status, and conducted morphological and phylogenetic studies” with "To clarify taxonomic status, morphological and phylogenetic studies were conducted”.

·         Page 2, line 77: Remove the "the" and the "plate".

·     Page 2, line 77: When describing how the monosporic isolation was done you say that the mycelium of the edge of the colony were picked and transferred to another PDA plate. Shouldn't it just be the tip of a hyphae, rather than the mycelium?

·   Page 2, line 78: Remove the "empty" and use “another Petri dishes containing PDA”.

·         Page 3, line 83: Replace "single colony” with “colonies”.

·         Page 4, line 145: Phyllosticta?

·    Page 4, lines 157-161: The text from “MCMC analyzes of...” to “…majority rule consensus tree (Fig. 1; first value: PP > 0.90 shown).” must be transferred to the methodology.

·      Page 4, line 163: The way you write (e.g.: "The 99 strains were classified into 91 species") suggests that you identified the 99 strains and that was not the case, so rewrite the text focusing on your isolates.

·         Page 4, line 172: three?

·         Page 20, line 324: Replace "the” with “its”.

·   Page 20, line 354: Insert a paragraph concluding the article, highlighting the contributions and importance of the study.

Kind regards,

Reviewer.

Comments on the Quality of English Language

I believe that an editing of English language is required.

Author Response

Dear Reviewer,

Thank you so much for your valuable suggestions. I accept all your suggestions and revised the content of the article according to your suggestions, and your modification of the format and grammar is of great help to me.

In addition to the errors and omissions in the format, I have greatly modified the relevant content of Apiospora indocalami sp. nov.. Accordingly, the description and discussion have also been revised, and a section on phylogenetic analysis has been added to the discussion.

Thank you so much again, and your prompt attention to this letter will be highly appreciated.

Best wishes,

Xinye Liu.

Round 2

Reviewer 2 Report

Comments and Suggestions for Authors

Dear authors, with respect to formal aspects, it is very important to be able to follow what you changed in the current version of the article and where you specifically modified the text according to reviewers suggestions. Track changes mode in Word documents is highly recomended or any other option like higlight the manuscript with colors, or add the specific changes in the response to revision.

In figure 1 I suggest to change the red color of the new species since it is commonly asociated with corrections. And in supplentary figures I suggest to clearly identify the new species in each tree to make it easier to follow.

With respect to the content it is very important that you added the single locus trees. However, after you added the single locus trees my concern about A. indoclami doesn´t disapear but it increased. Your philogenetic trees are not telling the same evolutionary history, are not congruent. It can invalidate the results of the concatenated matrix. From my point of view two options are possible: 1_ These loci suffer different evolutive pressure, in that case doesn´t follow the neutral theory of mutation proposed by Kimura and you are not allowed to combine the loci to tell the evolutive history of the species but you can tell the specific evolutive history of each clocus.

2_ You have "wrong" sequences (contaminations, bad quality, missidentified, wrongly trimmed...) causing the incongruence between trees.

I specifically checked the relative position of A. piptatheri which is close to A: indocalami in ITS, nested to A. gelatinousm in LSU, and almost as outgroup in TEF. These three positions are incompatible between them. For this reason I am almost sure that your single locus trees are not congruent.

I strongly suggest, a one by one revision of the sequences of each single locus tree, checking if there are any mistake with the sequences. If some sequence is incorrect or bad quality I suggest to avoid it. I am really concerned about the TEF single locus tree, which shows a very different topology, maybe you can think about it.

And to clear my concern about the different tree topologies, you should perform a congruence test between trees. (Adding the single locus trees is not enough when the topologies are different, in that case congruence should be tested).

Based on your data I don´t believe that A. indocalami is a different species than A. piptatheri, since they are not different in ITS, not clearly different in LSU, and the TEF is showing a not congruent topology with the other trees. In adition, the only morphological character proposed to discriminate between species are the conidia which shows overlaping size (including the error) between species. I suggest to avoid the description of A. indocalami as a new species until you will be able to provide stronger evidence of it.

  In the first row of revision I marked major revision, but I was close to accept after minor corrections. Now I am marking major revision, but I am closer to suggest rejection.

Author Response

Dear Reviewer,

Thanks for your suggestions.

We finally decided to identify A. indoclami sp. nov. as A. piptatheri (Pintos & P. Alvarado, 2019). And I changed the color of the new species in figure 1 and marked them in blue in figure s1-s4. The other modified parts of the text were highlighted in yellow.

As Apiospora and Arthrinium used to belong to the same taxa, I selected all the sequences in this paper based on recent publications and the latest Apiospora taxonomic literature to ensure that the sequences I used belonged to Apiospora and were more qualified in quality. I checked the TEF1-α sequences, especially those with topological anomalies, and tried to make sequence replacements, but after a search in NCBI, I was sorry to find that each of these species had only one TEF1-α sequence in the database that I used.

Since not all species both have LSU,TEF1-α, and TUB2 sequences, the number of sequences in each single locus trees is not the same, which will cause some differences in species position. Apiospora and Arthrinium are sister taxa, and phylogenetic analysis is the main basis for distinguishing them. However, the single locus tree contains too little genetic information, and the single locus sequence is affected by "wrong" as you mentioned, So the absence of information about other loci is not enough to determine whether the species we have found is Apiospora or Arthrinium. Therefore, I believe that single-locus tree can only play a certain auxiliary role and cannot determine the specific location of species. Only by adding more genetic information can the location range be gradually narrowed. Phylogenetic tree with multi-gene combination is more convincing than single-locus tree.

Thank you so much again and your prompt attention to this letter will be highly appreciated.

Best wishes,

Xinye Liu

Reviewer 3 Report

Comments and Suggestions for Authors

Dear Authors,

Analyzing the revised version of the manuscript, I found that you carried out the simplest and most typological revisions, however ignored several suggestions, which may be more laborious, but which will provide important improvements for this taxonomic study. In your response letter, you did not explain the reasons for not accepting these suggestions. So I will highlight these suggestions again and I would like arguments to be provided for those that are not accepted.

·         Include more Apiospora piptatheri sequences in phylogenetic analyzes to ensure that A. indocalami sp. nov. will not be positioned within the clade of A. piptatheri. Based on the phylogeny and also on the fact that the only morphological difference you pointed out between these species was tenths of a micrometer in conidial measurements, I consider it necessary to provide more robust evidence for the separation of these species. Then try this by including more A. piptatheri sequences in the tree and point out more morphological differences between the species in the Notes section after the description.

·         In the descriptions of the four species, try to characterize the conidiogenic cells in more detail. In the notes after the descriptions provide further morphological differences between the described species and their respective sister taxa. Note: I noticed that you included data about conidiogenic cells in the Notes section, but it was not clear whether the data included was about A. piptatheri or A. indocalami, so improve the writing of this part and remember that the characteristics for the species described must be presented in the description, the Notes section must have a more comparative writing style between the taxa.

·         Improve the legends of the five figures (See examples in taxonomy articles).

·         Page 2, line 59: insert a stop dot after "conducted" and capitalize the F of “four”.

·         Page 4, lines 162-163: The way you write (e.g.: "The 99 strains were classified into 91 species") suggests that you identified the 99 strains and that was not the case, so rewrite the text focusing on your isolates.

·         Page 21, line 375: Insert a paragraph concluding the article, highlighting the contributions and importance of the study.

Best regards,

The Reviewer.

Comments on the Quality of English Language

English language requires minor revisions.

Author Response

Dear Reviewer,

Thanks for your suggestions.

We finally decided to identify A. indoclami sp. nov. as A. piptatheri (Pintos & P. Alvarado, 2019). The other modified parts of the text were highlighted in yellow.

In the notes I added a morphological comparison between the new species and its sister taxa. And I tried to add details in the description, but because the form of Apiospora does not have more structure than other species such as Zygomyces, it did not add much content. I have read the relevant literature of Apiospora, and the morphological changes of Apiospora are few, and most species have similar morphology. The most intuitive evidence is still phylogenetic analysis.

At the end of the discussion section, I have added a paragraph that briefly summarizes the content of this article to avoid over-repetition.

About the legends of the five figures, I'm sorry I didn't quite understand what you meant. I checked the five figures and confirmed that the notes were added below each figure, and there were no missing elements such as ruler in the pictures. At the same time, I referred to the relevant literature of Apiospora and confirmed that there were no missing text contents in the notes. Do you want to express that the typography of figures and text needs to be adjusted?

Thank you so much again and your prompt attention to this letter will be highly appreciated.

Best wishes,

Xinye Liu

Round 3

Reviewer 2 Report

Comments and Suggestions for Authors

After the modifications that you did, my concerns have been minimized. The data used in the work has the bias that I exposed and you understand, and it is not your fault, since it is due to the limitations of the database. However this limitations limits the conclusions.

  In any case to my opinon should be important to add congruence test each time you are using different loci. But, I don´t consider it as a must for the other 3 species since the results looks strong enough.

Reviewer 3 Report

Comments and Suggestions for Authors

Dear authors, you have addressed my taxonomic concerns. The current version has been significantly improved and can be accepted.

With regards,

The Reviewer.

Comments on the Quality of English Language

The text still contains a few incomplete sentences (e.g.: In our phylogenetic analyses, 97 strains of Apiospora as a monophyletic clade) or words that could be improved (e.g.: eight strains conducted three new species lineages). For this reason, a general revision of the language would be welcome, but I no longer consider this an issue to be addressed to reviewers.